# Systematic review and meta-analysis of myopia prevalence in African school children

**Godwin Ovenseri-Ogbomo[1]**, **Uchechukwu L. Osuagwu**[2‡] *, **Bernadine N. Ekpenyong**[3‡], **Kingsley Agho**[4‡], **Edgar Ekure**[5], **Antor O. Ndep**[6], **Stephen Ocansey**[7], **Khathutshelo Percy Mashige**[8], **Kovin Shunmugan Naidoo**[8,9], **Kelechi C. Ogbuehi**[10]

1 Department of Optometry, Centre for Health Sciences, University of the Highlands and Islands, Inverness, United Kingdom, 2 Translational Health Research Institute (THRI), Western Sydney University, Campbelltown, New South Wales, Australia, 3 Epidemiology and Biostatistics Unit, Department of Public Health, University of Calabar, Calabar, Nigeria, 4 School of Science and Health, Western Sydney University, Campbelltown, New South Wales, Australia, 5 Salus University Pennsylvania, Pennsylvania, United States of America, 6 Department of Public Health, Faculty of Allied Medical Sciences, College of Medical Sciences, University of Calabar, Cross River State, Nigeria, 7 Department of Optometry and Vision Science, School of Allied Health Sciences, College of Health and Allied Sciences, University of Cape Coast, Cape Coast, Ghana, 8 African Vision Research Institute, Discipline of Optometry, University of KwaZulu-Natal, Westville Campus, Durban, South Africa, 9 School of Optometry and Vision Science, University of New South Wales, Sydney, New South Wales, 10 Department of Medicine, Dunedin School of Medicine, University of Otago, Dunedin, New Zealand

⊛ These authors contributed equally to this work.
‡ ULO, BNE and KA also contributed equally to this work.
* l.osuagwu@westernsydney.edu.au

**Data Availability Statement:** All relevant data are within the paper and its Supporting information files.

## Abstract

### Purpose

Increased prevalence of myopia is a major public health challenge worldwide, including in Africa. While previous studies have shown an increasing prevalence in Africa, there is no collective review of evidence on the magnitude of myopia in African school children. Hence, this study reviews the evidence and provides a meta-analysis of the prevalence of myopia in African school children.

### Methods

This review was conducted using the 2020 Preferred Reporting Items for Systematic reviews and Meta-Analyses (PRISMA) guidelines. Five computerized bibliographic databases, PUBMED, Scopus, Web of Science, ProQuest, and Africa Index Medicus were searched for published studies on the prevalence of myopia in Africa from 1 January 2000 to 18 August 2021. Studies were assessed for methodological quality. Data were gathered by gender, age and refraction technique and standardized to the definition of myopia as refractive error $\geq$ 0.50 diopter. A meta-analysis was conducted to estimate the prevalence. Significant heterogeneity was detected among the various studies ($I^2$ >50%), hence a random effect model was used, and sensitivity analysis was performed to examine the effects of outliers.

### Results

We included data from 24 quality assessed studies, covering 36,395 African children. The overall crude prevalence of myopia over the last two decades is 4.7% (95% CI, 3.9–5.7) in

**Funding:** The authors recieved no specific funding for this work.

**Competing interests:** The authors have declared that no competing interests exist.

African children. Although the prevalence of myopia was slightly higher in females (5.3%, 95%CI: 4.1, 6.5) than in males (3.7%, 95% CI, 2.6–4.7; p = 0.297) and higher in older [12–18 years 5.1% (95% CI, 3.8–6.3) than younger children (aged 5–11 years, 3.4%, 95% CI, 2.5–4.4; p = 0.091), the differences were not significant. There was a significantly lower prevalence of myopia with cycloplegic compared with non-cycloplegic refraction [4.2%, 95% CI: 3.3, 5.1 versus 6.4%, 95%CI: 4.4, 8.4; p = 0.046].

## Conclusions

Our results showed that myopia affects about one in twenty African schoolchildren, and it is overestimated in non-cycloplegic refraction. Clinical interventions to reduce the prevalence of myopia in the region should target females, and school children who are aged 12–18 years.

## Introduction

Uncorrected refractive error is the most common cause of visual impairment affecting an estimated one billion people globally [1]. Myopia is the most common refractive error and an important cause of ocular morbidity, particularly among school-aged children and young adults. Worldwide, myopia is reaching epidemic proportions linked to changing lifestyles and modern technology, particularly mobile devices [2]. Globally, myopia affected 22.9% of the world's population in 2000, with projections of an increase to 49.8% by 2050 affecting 4.8 billion people [2], representing a 117% increase over 50 years. According to a 2015 report, it was estimated that globally, about 1.89 billion people are myopic and 170 million have high myopia [3].

The reported prevalence of myopia in children aged 5–17 years ranges from 1.2% in Mechi Zone, Nepal, to 73.0% in South Korea [4, 5]. Over 15 years, the prevalence of myopia increased from 79.5% to 87.7% in Chinese high school children with an average age of 18.5 ± 0.7 years [6]. In South African school children aged 5–15 years, the reported prevalence of myopia was only 2.9% with retinoscopy and 4.0% using autorefraction [7]. The authors reported that this prevalence increased to 9.6% at age 15 years.

The increase in myopia prevalence will have a significant economic impact because of associated ocular health problems and visual impairment. Uncorrected myopia of between– 1.50 D and– 4.00 D can significantly affect vision to be regarded as a cause of moderate visual impairment and blindness, respectively [8]. Apart from its direct impact on visual impairment, high myopia [usually defined as a spherical equivalent $\geq$ 5.00D [4, 9, 10] of myopia, although the definitions used to grade myopia are variable] increases the risk of potentially blinding ocular pathologies such as retinal holes; retinal tears; retinal degeneration; retinal detachment; and myopic macular degeneration [3, 11]. Uncorrected myopia has huge social, economic, psychological and developmental implications [12]. The economic cost of refractive errors, including myopia, has been estimated to be approximately US$ 202 billion per annum [13], far exceeding that of other eye diseases.

The increasing prevalence of myopia has led to research in the study of the possible mechanism for myopia development, which has generated two broad themes: the role of nature (genetic influences) and nurture (environmental influences including lifestyle). Understanding the mechanism for the development of myopia is also being explored in the control of myopia.

Epidemiologic data from Southeast Asia has given credence to the association between near work and myopia, given the number of hours children from this region spend doing near work. Due to vast regional differences in culture, habits, socioeconomic status, educational levels and urbanization, there is uncertainty as to the exact magnitude of the myopia burden among African school–aged children and its trend over time [14].

In the last few decades, there has been a change in the lifestyle and behavior of people in Africa as a result of increasing urbanization [15]. Africa's urban population grew from 27 million in 1950 to 567 million in 2015 (a 2,000% increase), and now 50% of Africa's population live in one of the continent's 7,617 urban agglomerations of 10,000 or more inhabitants [16]. Consequently, more children and young adults in Africa are increasingly engaged in indoor and near work activities compared to earlier generations [17]. Children spend long hours doing schoolwork and, following the advent of technology, increasingly use mobile devices for gaming and other activities [18, 19]. These factors are thought to promote myopia development and/or progression [20–23].

Africa is the world's second largest and second most populous continent, after Asia, and it accounts for about 16% of the world's human population. While every global region will experience a decline in population by 2100, the African population is expected to triple. Africa's population is the youngest amongst all the continents, the median age in 2012 was 19.7 years compared to the global median of 30.4 years. This young population is an important asset for the continent's development. The challenges of the young population must be addressed in time as they constitute the bulk of the productive age of the economy. While rising myopia is a cause for global concern, it is not given due attention in Africa due to a lack of adequate prevalence data and prospective studies tracking the trend of myopia over decades [24]. Due to this, the representation of Africa is poor in studies predicting global trends of myopia [24]. The aim of this study was to systematically review the evidence and provide a meta–analysis of the prevalence of myopia in African school children which will address the knowledge gap and help understand the prevalence of myopia among this group in Africa.

## Materials and methods

This systematic review followed the framework of the Preferred Reporting Items for Systematic Reviews and Meta-Analyses (PRISMA. See Checklist in S1 File) [25]. The protocol for the review was registered with PROSPERO (#CRD42020187609).

### Search strategy and quality assessment

Two review team members (GO and BE) performed an independent systematic search and review of myopia in Africa using published data spanning the last two decades. Refractive error came into reckoning as a cause of visual impairment in the last two decades, following the change in the definition of visual impairment which was based on presenting visual acuity [26]. The search was conducted on 25th May and 18th August 2021. A third reviewer, KO, adjudicated where there were disagreements. The quality of each selected article was assessed using the checklist developed by Downs and Black [27] and each included article was assessed and scored on a 10-item scale (scoring is shown in S1 Table). The search was restricted to articles available online, articles mentioning prevalence of myopia in any region of Africa, and articles published in the English language. Searches included the following databases: Web of sciences, PubMed, ProQuest, MEDLINE, Scopus, and African Index Medicus from 1st of January 2000 to August 18, 2021.

We searched these databases using the following MeSH (Medical Subject Heading) terms and keywords: Refractive AND error AND Africa AND children AND prevalence. A number

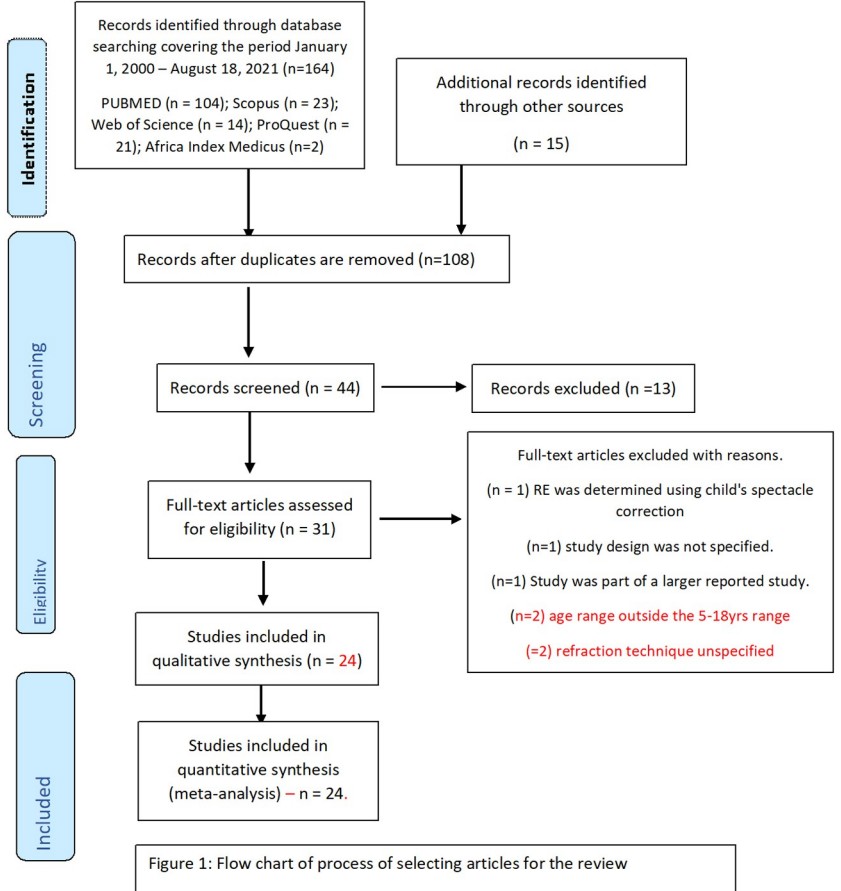

Fig 1. Flow chart of process of selecting articles for the review.

of iterations of these search terms were used, for example, "refractive error AND Africa AND children AND prevalence" or "refractive error AND Africa AND children". Further details about search strategy and MeSH terms are available in the (S2 File). A broader search also used terms such as epidemiology, myopia, and school children. We also identified and included relevant studies by manually searching through the reference lists of identified papers. The PRISMA flowchart presented in Fig 1 shows the process used for selecting articles.

## Inclusion and exclusion criteria

Studies published between 2000 and 2021, investigating the prevalence of refractive error in male and female school children aged 5 to 18 years of age were included in the review. Studies that employed an observational cross-sectional study design; had a clear description of the sampling technique; stated the method of measuring refractive error (cycloplegic or non-cycloplegic refraction), as well as objective or subjective refraction; stated the criteria for defining myopia (spherical equivalent $\geq$ 0.50 D of myopia [2, 28–30]; the study was either school–based or population–based; and were published in English language, were included in the review. The decision as to whether the articles met the inclusion criteria was made independently by the two reviewers (GO and BE) and where there was a disagreement, a third reviewer (KO) was consulted.

Studies where the criteria for defining myopia were not specified; the ages of the participants were either not specified or outside the age range specified for this review; or which reported findings from a hospital/clinic-based sample were excluded from the review.

## Data extraction

The data extracted from each article included the following: Authors; year of publication; country of study; study design; sample size; sampling technique; the age of study participants; criteria for defining myopia; method of refractive error assessment (cycloplegic vs non-cycloplegic); method of refractive error assessment (objective vs subjective); prevalence of myopia; and the proportion of refractive error due to myopia. Where the reported prevalence was not clearly defined, the corresponding author in the published article was contacted for clarification.

## Statistical methods

Meta–analysis was conducted using Stata version 14.0 (StataCorp, College Station, TX, USA). The syntax "metaprop" in Stata was used to generate forest plots and each forest plot showed the prevalence of myopia in school children, by gender, age and refraction technique in individual studies and its corresponding weight, as well as the pooled prevalence in each subset and its associated 95% confidence intervals (CI). A heterogeneity test obtained for the different studies showed a high level of inconsistency ($I^2 > 50\%$) thereby indicating the use of a random effect model in all the meta–analyses conducted. Sensitivity analysis was carried out by examining the effect of outliers, by employing similar method to that used by Patsopoulos et al. [31], which involves the process of comparing the pooled prevalence before and after eliminating one study at a time. The funnel plot was used to report the potential bias and small/large study effects and Begg's tests was used to assess asymmetry. The prevalence was subdivided into separate datasets based on overall prevalence, males or females, cycloplegic or non-cycloplegic refraction for a more detailed analysis of the prevalence of myopia. Also, to study a possible variation of the prevalence of myopia in terms of age, the age groups in the reported studies were divided into two categories: 5–11 years and 12–18 years. Their respective funnel plots are shown as (S3–S7 Files).

## Results

### Summary of included studies

From the described search strategy, a total of 164 potentially relevant titles/abstracts of articles were initially identified. Fig 1 presents the flowchart of the article screening and selection process. Following a quick inspection of identified studies and removal of duplicate articles, 44 relevant articles were assessed for eligibility. Using the pre–defined inclusion and exclusion criteria, 24 of 30 articles that underwent detailed review were eligible, and data from these studies were included in this study. A breakdown of the eligible studies as well as their quality assessment scores (maximum of 10) are presented in Table 1. S1 Table shows how the quality assessment scores were calculated.

The included studies comprised of the following: six (25.0%) studies from Ghana, four (16.7%) each from South Africa, and Nigeria, three from Ethiopia (12.5%), two (8.3%) from Burkina Faso, and one (4.2%) each from Sudan, Egypt, Equatorial Guinea, Somalia and Tunisia (Table 1). Of the reviewed articles, 84.2% (n = 21) were school–based, cross–sectional studies, two (8.3%) were population–based, cross–sectional studies, while one (4.2%) employed a cross–sectional study design but did not report whether it was school or population–based.

**Table 1. Characteristics of studies that reported the prevalence of myopia in school–aged children in Africa and were included in the meta–analysis.**

| First Author | Year of study | Study Country† | Age group (years) | Mean age (years) | Total Sample size | Cycloplegia | Objective refraction | Prevalence of myopia (%) | Common refractive error | Total Quality Assessment score |
|---|---|---|---|---|---|---|---|---|---|---|
| Atowa [32] | 2017 | Nigeria | 8–15 | 11.5 ± 2.3 | 1197 | Yes | Objective | 2.7 | | 10 |
| Wajuihian [33] | 2017 | South Africa | 13–18 | 15.8 ± 1.6 | 1586 | No | Objective | 7 | | 10 |
| Chebil [34] | 2016 | Tunisia | 6–14 | 10.1 ± 1.8 | 6192 | Yes | Objective | 3.71 | | 9 |
| Kedir [35] | 2014 | Ethiopia | 7–15 | Not reported | 570 | No | Subjective | 2.6 | | 10 |
| Soler [36] | 2015 | Equatorial Guinea | 6–16 | 10.8 ± 3.1 | 425 | Yes | Objective | 10.4 | | 8 |
| Kumah [37] | 2013 | Ghana | 12–15 | 13.8 | 2435 | Yes | Objective | 3.2 | | 10 |
| Mehari [38] | 2013 | Ethiopia | 7–18 | 13.1 ± 2.5 | 4238 | No | Objective | 6 | | 9 |
| Jimenez [39] | 2012 | Burkina Faso | 6–16 | 11.2 ± 2.4 | 315 | No | Objective | 2.5 | | 8 |
| Naidoo [7] | 2003 | South Africa | 5–15 | Not reported | 4890 | Yes | Objective | 2.9 | | 9 |
| Yamamah [40] | 2015 | Egypt | 6–17 | 10.7 ± 3.1 | 2070 | Yes | Objective | 3.1 | Astigmatism | 10 |
| Nartey [41] | 2016 | Ghana | 6–16 | 10.6 | 811 | No | Subjective | 4.6 | | 10 |
| Anera [42] | 2006 | Burkina Faso | 5–16 | 10.2 ± 2.2 | 388 | Yes | Objective | 0.5 | | 7 |
| Chukwuemeka [43] | 2015 | South Africa | 7–14 | 9.9 ± 2.2 | 421 | No | Objective | 18.7 | Astigmatism | 10 |
| **Alrasheed [44]** | 2016 | Sudan | 6–15 | 10.8 ± 2.8 | 1678 | Yes | Objective | 6.8 | Myopia | 10 |
| **Abdul-Kabir [45]** | 2016 | Ghana | 10–15 | Not reported | 208 | No | Objective | 22.6 | Myopia | 10 |
| **Ebri [46]** | 2019 | Nigeria | 10–18 | 13.3 ± 1.9 | 4241 | Yes | Objective | 4.8 | Astigmatism | 10 |
| **Ezinne [47]** | 2018 | Nigeria | 5–15 | 9.0 ± 2.5 | 998 | Yes | Objective | 4.5 | Myopia | 10 |
| **Nakua [48]** | 2015 | Ghana | 12–17 | Not reported | 504 | No | Objective | 2.18 | Astigmatism | 10 |
| **Ndou [49]** | 2014 | South Africa | 8–15 | 11.0 | 476 | No | Subjective | 2.94 | | 10 |
| **Alrasheed [50]** | 2020 | Somalia | 6–15 | 11.2 ± 2.5 | 1204 | No | Objective | 9.1 | Myopia | 10 |
| **Ovenseri-Ogbomo [51]** | 2010 | Ghana | 11–18 | 14.5 ± 1.5 | 595 | No | Subjective | 1.7 | Hyperopia | 9 |
| **Ovenseri-Ogbomo & Omuemu [52]‡** | 2010 | Ghana | 5–18 | 10.5 ± 3.4 | 953 | Yes | Objective | 14.1 | Myopia | 9 |
| **Assem [53]** | 2021 | Ethiopia | 6–18 | 12.0 ± 2.4 | 601 | Yes | Objective | 8.5 | | 10 |
| **Maduka-Okafor [54]** | 2021 | Nigeria | 5–15 | 10.5 ± 2.7 | 5723 | Yes | Objective | 2.7 | Myopia | 10 |

† = country the study was conducted;

‡ = authors provided data for only those aged 5–18 years.

### Method of measuring refractive error in African school–aged children

Of the reviewed studies, 13 (54.2%) performed cycloplegic refraction to determine the refractive error status of the children, while non-cycloplegic refraction was used in 11 (45.8%) of the studies. Regarding the technique used for refractive error measurement, over three–quarters of the studies (n = 20, 83.3%) performed objective refraction, with about one–sixth (n = 4, 16.7%) performing subjective refraction.

### Prevalence of myopia in African school–aged children

The number of children aged 5–18 years included in the study ranged from 208 for a study conducted in Ghana [45] to 6192 for another study conducted in Tunisia [34, 55]. The

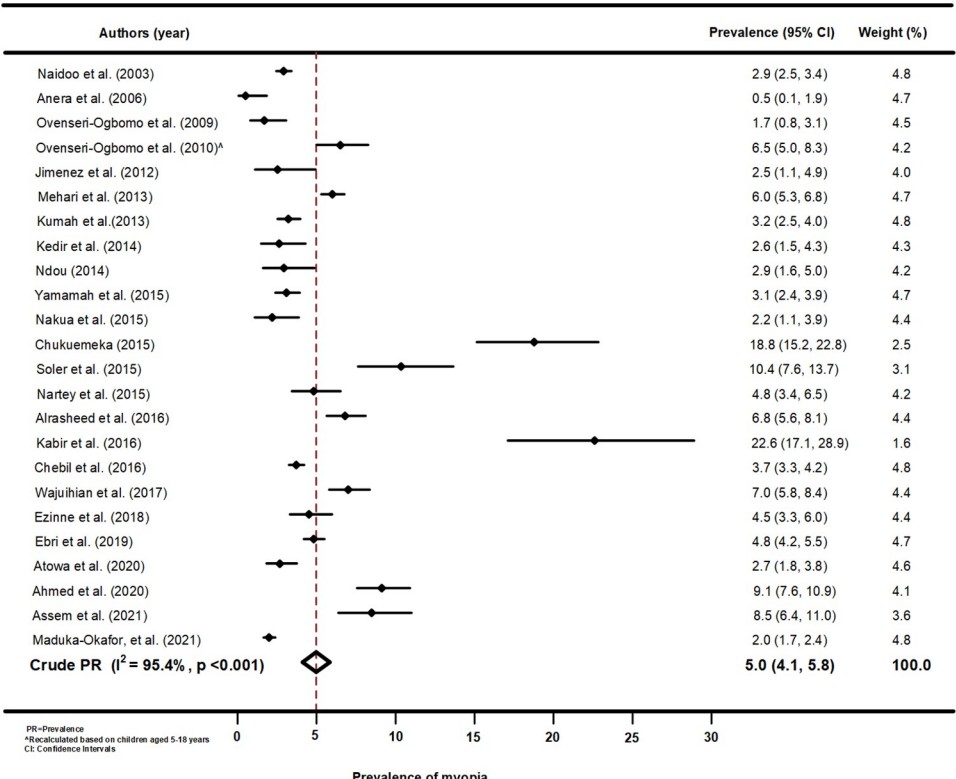

**Fig 2. Forest plot of myopia prevalence from the meta–analysis of African studies.**

prevalence of myopia reported in these studies ranged from 0.5% [42] to 10.4% [36, 52] with cycloplegic refraction. In studies where non–cycloplegic refraction was used to determine refractive error refraction in school children, the reported myopia prevalence ranged from 1.7% [51] to 22.6% [45].

## Meta-analysis of myopia prevalence in children ag 5–18 years in Africa (2000–2021)

**Myopia prevalence among school children in Africa.**　Fig 2 shows a forest plot of the prevalence of myopia among African school children aged 5–18 years. The pooled estimate of myopia in the African region was significant (5.0%, 95%CI: 4.1, 5.8; p<0.001) and about 37.5% of the studies (n = 9) reported significantly higher prevalence of myopia and 50% (n = 12) reporting significantly lower prevalence compared with the pooled estimate across Africa. The study by Abdul–Kabir found the highest prevalence (22.6%) of myopia among Ghanaian children (95%CI: 17.1, 28.9) [45], while Anera et al. found the lowest prevalence among children in Burkina Faso (0.5%, 95%CI: 0.1, 1.9) [42]. The pooled prevalence estimates of myopia was similar to the study by Ebri [46] and Ezinne [47] (4.8%, 95%CI:4.2, 5.5), both involving children from Nigeria [46, 47]. Funnel plots and using Begg's test for Myopia in Africa indicated homogeneity (S3 File) and meta–regression analysis of myopia by year of publication indicated that publication of year increased as the proportion of myopia decreased but this relationship was not statistically significant (p = 0.423, S7 File).

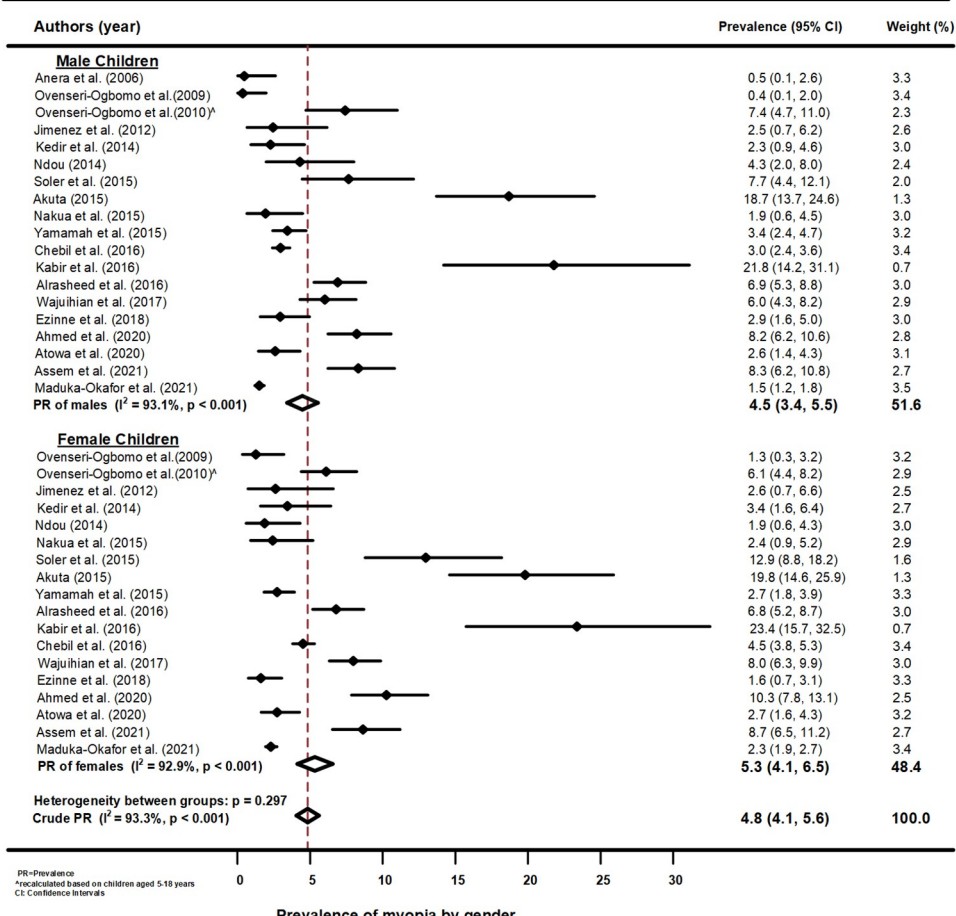

**Fig 3. Forest plot of myopia prevalence by gender from the meta-analysis of African studies.**

## Myopia prevalence by gender of the School children in Africa (2000–2021)

Fig 3 is a forest plot for prevalence of myopia by gender among school children aged 5–18 years in Africa. The prevalence estimates varied significantly between studies in both male and female children (p<0.001, per gender), and the overall pooled prevalence of myopia by gender was 4.8% (95%CI: 4.1, 5.6) and similar between male and female estimates (p = 0.297). Compared with the overall pooled estimate, the prevalence of myopia was slightly higher in male (4.5%, 95%CI: 3.4, 5.5) children than females (5.3%, 95%CI: 4.1, 6.5) but the difference was not significant as indicted by the overlapping of the CIs with that of the overall pooled estimate. Funnel plots and using Begg's test for Myopia by gender reported absence of publication biases (S4 File).

## Myopia prevalence by age group of the school children in Africa (2000–2021)

The forest plot of the prevalence of myopia in children aged 5–11 years and 12–18 years is presented in Fig 4. The pooled estimate of myopia in school children aged 5–11 years and 12–18 years was lower (3.7%, 95%CI 2.6, 4.7) and higher (5.8%, 95%CI 3.8, 6.3) respectively, than the pooled estimate but none was significant as they overlapped with the pooled estimate in Africa

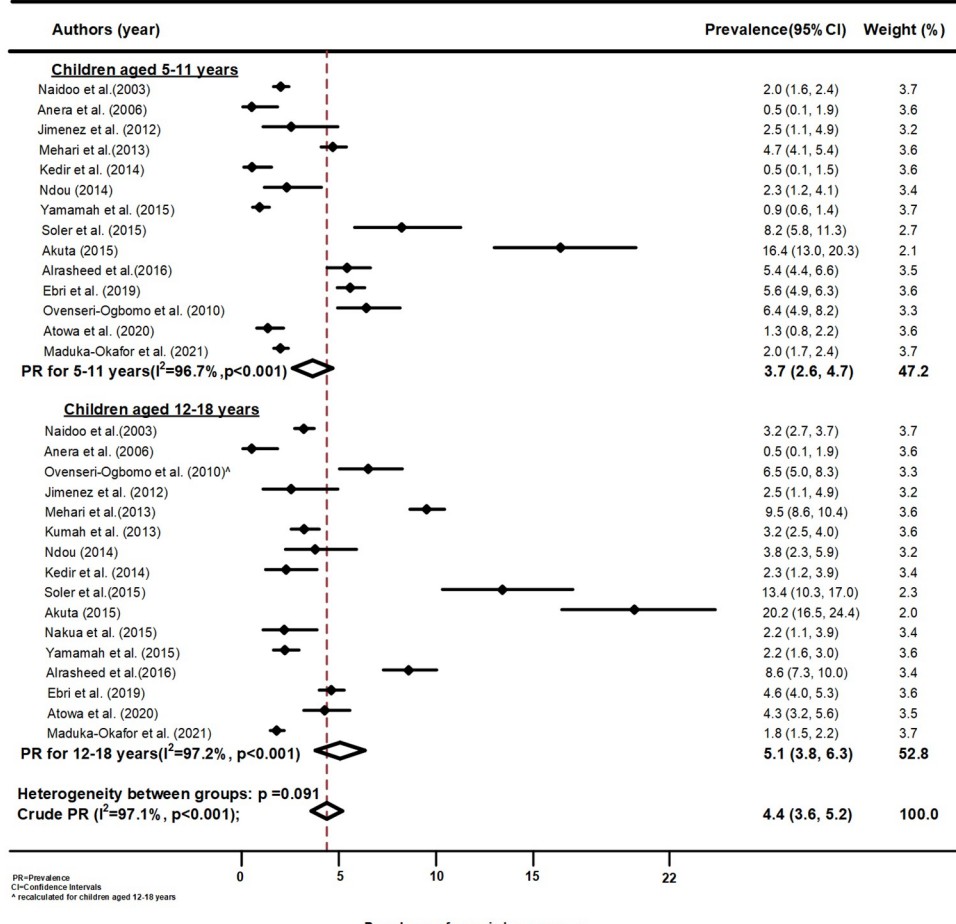

**Fig 4. Forest plot of myopia prevalence by age group across African studies.**

(4.4%, 95%CI 3.6, 5.2). The heterogeneity between the groups was approaching significant (p = 0.091) but older children had a higher prevalence of myopia than younger children. Among those aged 5–11 years, the highest significant prevalence was reported in a Ghanaian study (16.4%, 95%CI: 13.0, 20.3) and a study conducted in Equatorial Guinea (8.2%, 95%CI: 5.8, 11.3) while school children in Ethiopia (0.5%, 95%CI: 0.1, 1.5) had the lowest significant prevalence estimate of myopia. Among those aged 12–18 years, children in Ghana also showed the highest significant prevalence of myopia (20.2%, 95%CI: 16.5, 24.4), but the lowest prevalence was reported among School children in Burkina Faso (0.5%, 95%CI: 0.1, 1.9). The heterogeneity of these studies by age as subgroups analysis were low (S5 File).

## Myopia prevalence by mode of refraction among school children in Africa (2000–2021)

The forest plot displayed in Fig 5 shows the pooled estimate of myopia prevalence among school children in Africa. Using cycloplegic refraction, studies have reported significantly lower prevalence estimates of myopia among school children in Africa compared with those that used non–cycloplegic refraction (4.2%, 95%CI: 3.3, 5.1 versus 6.4%, 95%CI: 4.4, 8.4; p = 0.046). From the plot, it can be seen that studies that used non cycloplegic technique to determine refraction had greater variabilities in the reported myopia prevalence (ranging from

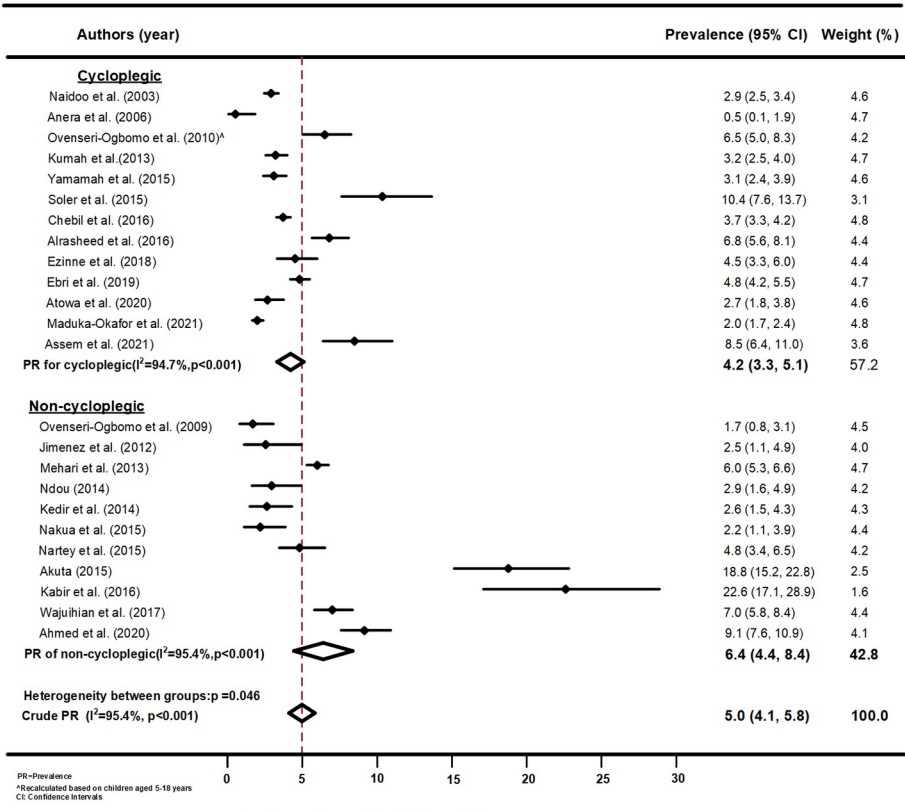

**Fig 5. Forest plot of myopia prevalence by refraction technique among school children in Africa.**

1.7 to 22.6%), but those that performed cycloplegic refraction had smaller between study variability in the reported prevalence of myopia (range from 0.5 to 10.4%). Funnel plots and the Begg's test for Myopia by refraction technique shown in S6 and S7 Files, respectively, found no publication biases.

## Discussion

### Prevalence of myopia

The present study provided recent estimates of the myopia prevalence in African children using data from twenty eight studies conducted over two decades. The prevalence of myopia defined as SER ≥ 0.50D of myopia in school children across African countries was 4.7% (95% CI, 3.9, 5.7%) and there were wide variations within and between African countries. A significantly higher prevalence rate was observed in Ghana [45] and South Africa [43], with significantly lower rates in Burkina Faso [42] and Ethiopa [56]. In some countries like Ghana, the variation in the reported prevalence of myopia between studies reached 21% [37, 41, 45, 48, 51, 52]. Although the regional variations in myopia prevalence found in this study are consistent with the statement of Foster and Jiang who remarked that "Considerable regional difference exists from country to country even within the same geographical area" [57], it remains unclear why these variations exist. While the criteria for defining refractive error is often cited as the reason for the variation in the prevalence of refractive errors, including myopia, between

studies, this may not be the case in our study because only studies that defined myopia as spherical equivalent of $\geq 0.50$ D were included.

The overall low prevalence of myopia found across Africa is consistent with other studies that reported lower myopia prevalence in African children compared with Asian children [5, 58]. It is instructive to note that in four of the studies that were included in the current review [36, 43, 45, 52], the reported prevalence of myopia was greater than 10%. Of these, two studies [36, 52] used cycloplegic refraction, which is thought to more accurately estimate the prevalence of myopia [59]. The lower prevalence of myopia in Africa compared with the other regions may be related to the differences in genetic predisposition to myopia development, and to culture [60–62]. Although the role of genetics in the development and progression of myopia is reported to be small [12], it is believed to have a role in an individual's susceptibility to environmental risk factors for myopia [63]. In addition, several studies have shown the major involvement of environmental factors such as near work (writing, reading, and working on a computer) in myopia development [60, 63]. In many African countries, children do not start education and learning at the same early age as in other countries of Asia. African children are therefore exposed to less near work and are more involved with outdoor activities, resulting in less risk of developing myopia compared with their Asian counterparts. This assertion is supported by the fact that in 2010, the pre-primary school enrolment rate in the most populous country in Africa (Nigeria) was 41.83% compared to 89.12% in 2012 in China (the most populous country in Asia) [64]. We acknowledge that a recent investigation [65] has shown that more precise objective measures are required to make definitive conclusions about the relationship between myopia and near work.

Notwithstanding the relatively low prevalence of myopia found among African children, there is a need to monitor myopia prevalence among children in this region given the increasing access to, and use of, mobile devices among African population [19], including children. This is important considering the reported higher increase in the prevalence of myopia in black children living in Africa (2.8% to 5.5%) compared with other black children not living in Africa (4.8% to 19.9%) after 10 years [58]. It is assumed that black children not in Africa may have more access and exposure to near work, including mobile devices, and less outdoor activities than their counterparts in Africa.

### Age and gender-based differences in myopia prevalence

There was a 34.6% increase in the prevalence of myopia between the age groups with the older age group having a higher prevalence of 5.2%. The slightly higher prevalence of myopia between the two age groups shows there is a tendency for myopia prevalence to increase with age which is consistent with previous studies from elsewhere [58, 66, 67]. This increase in myopia prevalence is thought to be associated with the increasing growth of the eyeball. Although the pooled prevalence of myopia in female children was slightly higher than in male children (4.7 versus 3.7%), the difference did not reach statistical significance. The influence of gender on the prevalence of myopia has not been unequivocal in the literature [68–72] with some suggesting that the slightly higher prevalence in females may be related to the different ages of onset of puberty between boys and girls [73]. Other factors that could account for the reported apparent higher prevalence of myopia in girls include limited outdoor activity time than boys [74].

### Prevalence of myopia by refraction technique (cycloplegic and non-cycloplegic)

The present study demonstrated that cycloplegic refraction resulted in significantly lower estimates of myopia prevalence than non-cycloplegic refraction, which was consistent with

previous studies [75–78]. It has been reported that non-cycloplegic refraction overestimates the prevalence of myopia, yields a non-reliable measurement of association of myopia risk factors [59, 76], and hence cycloplegic refraction is regarded as the gold standard for measuring myopia [59]. Over half of the studies in this review utilised cycloplegic refraction, which is particularly important in this age group where the difference between the cycloplegic and non-cycloplegic refraction is quite high [77, 78]. The fact that non-cycloplegic refraction often results in overestimation of myopia may have, in part, accounted for the high prevalence reported in one study from Ghana [45]. Furthermore, we have demonstrated that cycloplegic refraction results in a lower variability of measured refractive error than non–cycloplegic refraction (see Fig 5), which may reflect the variable accommodative state during the refraction of children of different ages. This finding underscores the need to appropriately control accommodation when performing refraction especially in young children who have a higher amplitude of accommodation and in whom accommodation is more active.

## Implications of the study

This is the first systematic review and meta-analysis to estimate the prevalence of myopia among school children in Africa and its variation with age, gender and refraction technique. As previously reported, the prevalence of myopia in Africa appears low compared to other regions such as South East Asia. This study also provides baseline data for comparison and future prevalence studies to establish a trend in myopia epidemiology in this population. A further remarkable finding in this review is the demonstration that non–cycloplegic refraction overestimated the prevalence of myopia and results in more variable estimates of refractive errors compared with cycloplegic refraction. The interpretation of myopia prevalence data obtained from non–cycloplegic refraction may be potentially misleading to researchers and policymakers. As a result, it is recommended that cycloplegic refraction be used in all studies investigating the prevalence of myopia in children.

## Strengths and limitations of the review

This review has certain limitations. Firstly, this review did not investigate the trend in the prevalence of myopia among school children in Africa due to the limited number of studies. Secondly, the selection of English-only studies likely biased the results towards studies in Anglophone countries or countries where the findings were reported in English. Thirdly, the current review did not explore the various factors influencing the epidemiology of myopia in this population. Despite these limitations, a major strength of this study is the selection of studies that used a uniform definition of myopia (i.e. $\geq$ 0.50DS of myopia) which allowed for a better comparison in the reported prevalence of myopia. In addition, the study excluded studies that were conducted in unselected groups such as hospital-based studies and studies that did not report any evidence of sampling in the study. In addition, the selected studies were evaluated for robustness in the study designs employed in each study.

## Conclusions

In summary, this systematic review and meta–analysis have shown that the prevalence of myopia among schoolchildren in Africa is lower than other regions of the world. The use of non–cycloplegic refraction for estimation of myopia prevalence can be misleading as it returns higher and more variable prevalence estimates. There is a need to monitor the trend of myopia as more children in this region are increasingly being exposed to identified risk factors for myopia development including access to mobile devices, increased near work, increased online or remote learning, and limited time outdoors. Future studies are needed to understand the

role of ethnicity on the myopia prevalence in Africa as the inclusion and comparison of the different ethnicities (Black vs White vs Asian) in the same region would add useful information about whether significant differences in the prevalence of myopia among different ethnicity in Africa exists.

## Supporting information

**S1 Table. Quality assessment of full-text articles included in review.**
(DOCX)

**S1 File. PRISMA 2020 checklist.**
(DOCX)

**S2 File. Search terms for refractive error Africa children prevalence filters (2000–2021).**
(DOCX)

**S3 File. Funnel plots and 95% confidence intervals of Myopia.**
(DOCX)

**S4 File. Funnel plots and 95% confidence intervals of Myopia by gender.**
(DOCX)

**S5 File. Funnel plots and 95% confidence intervals of Myopia by age in categories.**
(DOCX)

**S6 File. Funnel plots and 95% confidence intervals of Myopia by refraction technique.**
(DOCX)

**S7 File. A meta-regression analysis of Myopia by year of publication.** The vertical axis is the log proportion of Myopia, and the horizontal axis represents year of publication. Each dark dot represented one selected study, and the size of each dark dots corresponds to the weight assigned to each study. Given the slope of the regression line has descending slightly in this figure, this could be interpreted as publication of year increased, the proportion of myopia decreased and, this relationship did not differ statistically (p = 0.5512).
(DOCX)

**S8 File. Data used in the analysis.**
(XLSX)

## Acknowledgments

The authors acknowledge the guidance of late Prof Alabi, O Oduntan during data collection.

## Author Contributions

**Conceptualization:** Godwin Ovenseri-Ogbomo, Uchechukwu L. Osuagwu, Bernadine N. Ekpenyong, Kingsley Agho, Edgar Ekure, Antor O. Ndep, Khathutshelo Percy Mashige, Kovin Shunmugan Naidoo, Kelechi C. Ogbuehi.

**Data curation:** Godwin Ovenseri-Ogbomo, Kelechi C. Ogbuehi.

**Formal analysis:** Uchechukwu L. Osuagwu, Kingsley Agho.

**Investigation:** Godwin Ovenseri-Ogbomo, Uchechukwu L. Osuagwu, Bernadine N. Ekpenyong, Kingsley Agho, Edgar Ekure, Stephen Ocansey, Khathutshelo Percy Mashige, Kelechi C. Ogbuehi.

**Methodology:** Uchechukwu L. Osuagwu, Bernadine N. Ekpenyong, Kingsley Agho, Edgar Ekure, Antor O. Ndep, Stephen Ocansey, Kovin Shunmugan Naidoo, Kelechi C. Ogbuehi.

**Project administration:** Godwin Ovenseri-Ogbomo, Uchechukwu L. Osuagwu.

**Resources:** Khathutshelo Percy Mashige.

**Software:** Kingsley Agho.

**Supervision:** Kovin Shunmugan Naidoo, Kelechi C. Ogbuehi.

**Validation:** Stephen Ocansey.

**Writing – original draft:** Godwin Ovenseri-Ogbomo, Uchechukwu L. Osuagwu, Bernadine N. Ekpenyong, Khathutshelo Percy Mashige, Kelechi C. Ogbuehi.

**Writing – review & editing:** Godwin Ovenseri-Ogbomo, Uchechukwu L. Osuagwu, Bernadine N. Ekpenyong, Kingsley Agho, Edgar Ekure, Antor O. Ndep, Stephen Ocansey, Khathutshelo Percy Mashige, Kovin Shunmugan Naidoo, Kelechi C. Ogbuehi.

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
