## [Decision Letter · Decision Letter 0]

13 Dec 2021

PONE-D-21-28841Systematic Review and Meta-analysis of Myopia prevalence in African School children.PLOS ONE

Dear Dr. Osuagwu,

Thank you for submitting your manuscript to PLOS ONE. After careful consideration, we feel that it has merit but does not fully meet PLOS ONE’s publication criteria as it currently stands. Therefore, we invite you to submit a revised version of the manuscript that addresses the points raised during the review process.

We look forward to receiving your revised manuscript.

Kind regards,

Aleksandra Barac

Academic Editor

PLOS ONE

Journal Requirements:

Reviewers' comments:

Reviewer's Responses to Questions

**Comments to the Author**

1. Is the manuscript technically sound, and do the data support the conclusions?

Reviewer #1: Yes

Reviewer #2: Yes

2. Has the statistical analysis been performed appropriately and rigorously? 

Reviewer #1: Yes

Reviewer #2: Yes

3. Have the authors made all data underlying the findings in their manuscript fully available?

Reviewer #1: Yes

Reviewer #2: No

4. Is the manuscript presented in an intelligible fashion and written in standard English?

Reviewer #1: Yes

Reviewer #2: Yes

5. Review Comments to the Author

Reviewer #1: The authors conducted a review and meta-analysis of articles on the prevalence of myopia in African children.

This study follows the recommendations for this type of review.

Several points of detail should be reported

1 ° In the inclusion criteria, the authors report having excluded studies in which the ages of the participants were either not specified or outside the age range specified. But they did not clearly define the age ranges of this review themselves.

2 ° Two articles have been included but pose a problem in my opinion.

- They did not report whether it was school- or population-based. The inclusion / exclusion criteria are not clear at this level

- They did not specify the method used to determine the refractive error. However, it is clearly specified in the inclusion criteria "stated the method of measuring refractive error - cycloplegic or non-cycloplegic refraction, as well as objective or subjective refraction"

I think we should exclude these articles or change the inclusion criteria

3 ° in the table, in addition to the age limits, the median or average of the ages must be included in each article. Moreover, the authors specify it for an article: In another study (43) however, the children were aged 4 - 24 years but with a mean age of 12 years.

4 ° in the discussion, when the authors evoke the fact that fewer children await early education and learning in many African countries, compared with Asian countries, means that the children do less near work and are more involved with outdoor tasks, nuances must be made.

In a meta-analysis, Gajjar (Acta ophtahlmol 2021) show that the role of near vision is still questionable and that the study of the literature does not allow a conclusion. On the other hand, Tang Y (J Glob Health. 2021) shows the existence of a difference in the prevalence of myopia in China depending on whether the children live in the city or in the countryside.

5° The authors said that "he apparent higher prevalence of myopia in girls may be due to girls having ... shorter axial length than boys". That surprising !!!

Reviewer #2: This is a good Meta-analysis regarding the myopia prevalence in Africa

it is good structured and well-written; however, it would be better if you add a figure showing prevalence of myopia by ethnicity (black vs white vs asian in the same region) to show if it affects the prevalence of myopia or not

6. PLOS authors have the option to publish the peer review history of their article (what does this mean?). If published, this will include your full peer review and any attached files.

Reviewer #1: No

Reviewer #2: No

---

## [Author Response · Author response to Decision Letter 0]

13 Jan 2022

Response to Reviewers comments

Dear Aleksandra Barac

Thanks for the very useful comments which has strengthened our manuscript. We have revised the article according to the suggested comments. We have provided a point-by-point response to all reviewers comments for clarity.

The changes made in the revised manuscript and supplementary files were highlighted using red font for easy identification.

Journal Requirements:

Response: Done

Response: Done

Comments to the Author

1. Is the manuscript technically sound, and do the data support the conclusions?

Reviewer #1: Yes

Reviewer #2: Yes

2. Has the statistical analysis been performed appropriately and rigorously? 

Reviewer #1: Yes

Reviewer #2: Yes

3. Have the authors made all data underlying the findings in their manuscript fully available?

Reviewer #1: Yes

Reviewer #2: No

Response: We have included the study data used in the analysis as a spread sheet inline with PlosOne policy

4. Is the manuscript presented in an intelligible fashion and written in standard English?

Reviewer #1: Yes

Reviewer #2: Yes

5. Review Comments to the Author

Reviewer #1: 

The authors conducted a review and meta-analysis of articles on the prevalence of myopia in African children.

This study follows the recommendations for this type of review.

Several points of detail should be reported

1 ° In the inclusion criteria, the authors report having excluded studies in which the ages of the participants were either not specified or outside the age range specified. But they did not clearly define the age ranges of this review themselves. 

Response: Agreed and we have excluded the 4–24year-old range study (Yareed et al) and the 5-19 year study (Ovenseri-Ogbomo et al) as they do not meet our stipulated inclusion criteria of 5-18 year. 

2 ° Two articles have been included but pose a problem in my opinion.

- They did not report whether it was school- or population-based. The inclusion / exclusion criteria are not clear at this level. They did not specify the method used to determine the refractive error. However, it is clearly specified in the inclusion criteria "stated the method of measuring refractive error - cycloplegic or non-cycloplegic refraction, as well as objective or subjective refraction" 

Response: The inclusion and exclusion criteria were made clearer and as suggested, we excluded these studies as the two stipulated criteria are not specified [Rushood (39) and Woldeamanuel (47)] 

3 ° in the table, in addition to the age limits, the median or average of the ages must be included in each article. Moreover, the authors specify it for an article: In another study (43) however, the children were aged 4 - 24 years but with a mean age of 12 years. 

Response: We have included the mean age in Table 1 and the study with age range 4-24years was excluded based on the exclusion criteria.

4 ° in the discussion, when the authors evoke the fact that fewer children await early education and learning in many African countries, compared with Asian countries, means that the children do less near work and are more involved with outdoor tasks, nuances must be made. 

Response: In a meta-analysis, Gajjar (Acta ophthalmol 2021) showed that the role of near vision is still questionable and that the study of the literature does not allow a conclusion. On the other hand, Tang Y (J Glob Health. 2021) showed the existence of a difference in the prevalence of myopia in China depending on whether the children live in the city or in the countryside. However, we agree with the reviewer and have made the following revision in the discussion section: 

In addition, several studies have shown the major involvement of environmental factors such as near work (writing, reading, and working on a computer) in myopia development(62, 65). In many African countries, children do not start education and learning at the same early age as in other countries of Asia. African children are therefore exposed to less near work and are more involved with outdoor activities, resulting in less risk of developing myopia compared with their Asian counterparts. This assertion is supported by the fact that in 2010, the pre-primary school enrolment rate in the most populous country in Africa (Nigeria) was 41.83% compared to 89.12% in 2012 in China (the most populous country in Asia) (66). We acknowledge that a recent investigation(67) has shown that more precise objective measures are required to make definitive conclusions about the relationship between myopia and near work. 

5° The authors said that "he apparent higher prevalence of myopia in girls may be due to girls having ... shorter axial length than boys". That surprising !!! 

Response: Zadnik et al study was referring to a specific context in their study, where they found that girls tended to have steeper corneas, stronger crystalline lenses, and shorter eyes/axial length than boys. These findings are specific to their study and cannot be used to explain any result where a higher prevalence of myopia in girls is found. For example, we know that shorter axial length is generally associated with hyperopia and not myopia. 

However, the new analysis after removing the 4 studies, showed no statistically significant difference in myopia prevalence between gender. Therefore, we have removed this statement and the revised section now reads: 

The influence of gender on the prevalence of myopia has not been unequivocal in the literature (70-74) with some suggesting that the slightly higher prevalence in females may be related to the different ages of onset of puberty between boys and girls (75). Other factors that could account for the reported apparent higher prevalence of myopia in girls include limited outdoor activity time than boys (76).

Reviewer #2

This is a good Meta-analysis regarding the myopia prevalence in Africa. It is good structured and well-written; however, it would be better if you add a figure showing prevalence of myopia by ethnicity (black vs white vs asian in the same region) to show if it affects the prevalence of myopia or not 

Response: Thanks for the suggestion. Although the inclusion and comparison of the different ethnicities (Black vs White vs Asian) in the same region would add useful information about the differences in the prevalence of myopia between ethnic groups in Africa, studies that have been conducted in Africa did not specify the different ethnicities. However, we think there is need for such comparison between black vs white vs Asian and this could be another research interest with a different research aim for another manuscript. We have suggested this in the conclusion for future study direction. The section now reads: 

Future studies are needed to understand the role of ethnicity on the myopia prevalence in Africa as the inclusion and comparison of the different ethnicities (Black vs White vs Asian) in the same region would add useful information about whether significant differences in the prevalence of myopia among different ethnicity in Africa exists. 

6. PLOS authors have the option to publish the peer review history of their article (what does this mean?). If published, this will include your full peer review and any attached files.

Do you want your identity to be public for this peer review? For information about this choice, including consent withdrawal, please see our Privacy Policy.

Reviewer #1: No

Reviewer #2: No

Response. Thanks for your comments

---

## [Editor Report · Decision Letter 1]

17 Jan 2022

Systematic Review and Meta-analysis of Myopia prevalence in African School children.

PONE-D-21-28841R1

Dear Dr. Osuagwu,

We’re pleased to inform you that your manuscript has been judged scientifically suitable for publication and will be formally accepted for publication once it meets all outstanding technical requirements.

Kind regards,

Aleksandra Barac

Academic Editor

PLOS ONE

---

## [Editor Report · Acceptance letter]

24 Jan 2022

PONE-D-21-28841R1 

Systematic Review and Meta-analysis of Myopia prevalence in African School children. 

Dear Dr. Osuagwu:

I'm pleased to inform you that your manuscript has been deemed suitable for publication in PLOS ONE. Congratulations! Your manuscript is now with our production department. 

Kind regards, 

on behalf of

Dr. Aleksandra Barac 

Academic Editor

PLOS ONE